# Convex Polytope Trees

**Mohammadreza Armandpour**
Department of Statistics
Texas A&M University
armand@stat.tamu.edu

**Ali Sadeghian**
Department of Computer Science
University of Florida
asadeghian@ufl.edu

**Mingyuan Zhou**
McCombs School of Business
The University of Texas at Austin
mingyuan.zhou@mccombs.utexas.edu

## Abstract

A decision tree is commonly restricted to use a single hyperplane to split the covariate space at each of its internal nodes. It often requires a large number of nodes to achieve high accuracy. In this paper, we propose convex polytope trees (CPT) to expand the family of decision trees by an interpretable generalization of their decision boundary. The splitting function at each node of CPT is based on the logical disjunction of a community of differently weighted probabilistic linear decision-makers, which also geometrically corresponds to a convex polytope in the covariate space. We use a nonparametric Bayesian prior at each node to infer the community's size, encouraging simpler decision boundaries by shrinking the number of polytope facets. We develop a greedy method to efficiently construct CPT and scalable end-to-end training algorithms for the tree parameters when the tree structure is given. We empirically demonstrate the efficiency of CPT over existing state-of-the-art decision trees in several real-world classification and regression tasks from diverse domains.

## 1 Introduction

Decision trees [8] are highly interpretable models, which make them favorable in high-stakes domains such as medicine [32, 43] and criminal justice [6]. They are also resistant, if not completely immune, to the inclusion of many irrelevant predictor variables. However, trees usually do not have high accuracy, which somewhat limits their use in practice. Current main approaches to improve the performance of decision trees are making large trees or using ensemble methods [11, 12, 18, 53], such as bagging [7] and boosting [13, 16], which come with the price of harming model interpretability. There is a trade-off challenge between the accuracy and interpretability of a decision tree.

Prior work has attempted to address the aforementioned challenge and improve the performance of trees by introducing oblique tree models [19, 34]. These families of models are generalizations of classical trees, where the decision boundaries are hyperplanes that are not constrained to be axis-parallel and can have an arbitrary orientation. This change in the decision boundaries has been shown to reduce the size of trees. However, the tree size often remains too large in real datasets to make it amenable to interpretation. There has been an extensive body of research to improve the training of the oblique trees and enhance their performance [5, 9, 28, 46], yet their large size remains a challenge.

In this paper, we propose convex polytope decision trees (CPT) to expand the class of oblique trees by extending hyperplane cuts to more flexible geometric shapes. To be more specific, the decision

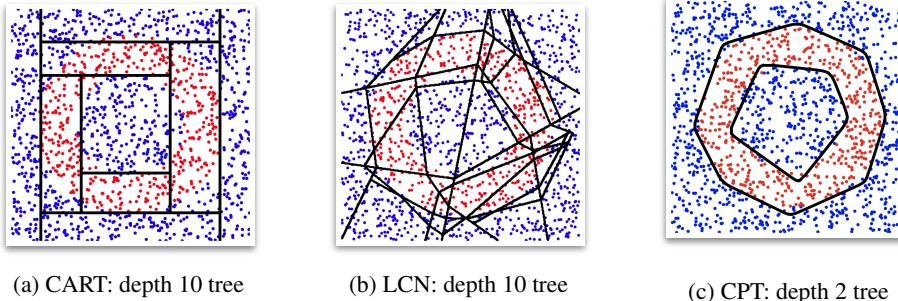

| (a) CART: depth 10 tree | (b) LCN: depth 10 tree | (c) CPT: depth 2 tree |

Figure 1: A visual comparison of the decision boundaries of CART, LCN and CPT on the synthetic 2D dataset. As demonstrated by plots (a) and (b), CART and LCN need to partition the space into many subspaces to achieve reasonable performance. This results in a multitude of smaller partitions which render the model harder to interpret. By contrast, as shown in plot (c), CPT successfully partitions the data with a much simpler scheme (depth 2).

boundaries induced by each internal node of CPT are based on noisy-OR [36] of multiple linear classifiers. And since noisy-OR has been widely accepted as an interpretable Bayesian model [37], our generalization keeps the interpretability of oblique trees intact. Furthermore, CPT's decision boundaries geometrically resemble a convex polytope (*i.e.*, high dimensional convex polygon). Therefore, the decisions at each node have both logical and geometrical interpretation. We use the gamma process [14, 25, 52, 51], a nonparametric Bayesian prior, to infer the number of polytope facets adaptively at each internal tree node and regularize the capacity of the proposed CPT. A realization of the gamma process consists of countably infinite atoms, each of which is used to represent a weighted hyperplane of a convex polytope. The shrinkage property of the gamma process helps us to encourage having simpler decision boundaries, therefore help resist overfitting and improve interpretability. Figure 1 provides an illustrative example of our model for a toy data.

Note, our main goal for proposing CPT is not to improve interpretability by having a smaller number of leaves rather it is pushing accuracy limits of decsion trees methods while staying in the class of interpretable models. Neural networks, with more than two layers, are often considered black-box models (not in the class of interpretable models). However, even with larger depth, lots of leaves, or complex (but interpretable) decision boundaries, decision trees would still remain in the class of interpretable models. Our proposed CPT offers higher accuracy than previously studied tree models and remains interpretable (noisy-OR decision boundaries are widely considered interpretable). While it is hard to objectively compare the interpretability ("Interpretability is a domain-specific notion so there cannot be an all-purpose definition" [38]) of convex polytope trees vs. oblique trees, they offer an alternative where the ease of interpretability of the decision boundaries at each node can be traded for shallower trees with significantly fewer leaves. Choosing one over the other is often application dependent, with the added advantage that convex polytope trees often offer higher accuracy.

The training of CPT, like that of oblique trees, is a challenging task because it requires learning both the structure (*i.e.*, the topology of the tree and the cut-off for the decision boundaries) and the parameters (*i.e.*, parameters of noisy-OR). The structure is a discrete optimization problem, involving the search over a potentially large problem space. In this work, we present two fully differentiable approaches for learning CPT models, one based on mutual information maximization, applicable for both binary and multi-class classification, and the other based on variance minimization, applicable for regression. The differentiable training allows one to use modern stochastic gradient descent (SGD) based programming frameworks and optimization methods for learning the proposed decision trees for both classification and regression.

Experimentally, we compare the performance of CPT to state-of-the-art decision tree algorithms [9, 27] on a variety of representative tasks in both regression and classification. We experiment with several real-world datasets from diverse domains, such as computer vision, tabular data, and chemical property data. Experiments demonstrate that CPT outperforms state-of-the-art methods with higher accuracy and smaller size.

Our main contributions include: 1) We propose an interpretable generalization to the family of oblique decision trees models; 2) We regularize the expressive power of CPT, using a nonparametric Bayesian

shrinkage prior for each node split function; 3) We provide two scalable and differentiable ways of learning CPT models, one for classification and the other for regression, which efficiently search for the optimal tree; 4) We experimentally evaluate CPT on several different types of predictive tasks, illustrating that this new approach outperforms the prior work in having higher accuracy achieved with a smaller size.

## 2 Related work

Most decision tree literature focuses on training a single or an ensemble of trees [18] instead of making decision boundaries more flexible. One reason for this lack of research is the increase in the problem's computational complexity, even for simple generalizations of the decision boundary. For example, with $N$ as the number of data points and $d$ as the dimension of the input space, generalizing the coordinate-wise to an oblique hyperplane cut, increases the number of possible splits of data points from $Nd$ to $\sum_{i=0}^{d} \binom{N}{i}$ just for a single node [44].

To have more flexible decision boundaries, some methods perform hyperplane cuts in an extended feature space, created by concatenating the original and newly generated features [1]. These new features are engineered or kernel-based and not designed for interpretability. Therefore unlike CPT, they are not an interpretable extension of the decision tree. We follow this section with a literature review of the recent training algorithms for axis-aligned and oblique trees.

Conventional methods for decision tree induction are greedy, where they grow the tree nodes one at a time. The greedy construction of oblique trees can be done by using coordinate descent to learn the parameters of each split [34], or by a projection of the feature space to a lower dimension then using coordinate-cut [33, 46]. However, the greedy procedure often leads to sub-optimal trees.

There have been several attempts to non-greedy optimization, which rely on either fuzzy or probabilistic split functions [22, 26, 40, 41]. The probabilistic trees are sometimes referred to as soft decision trees [17, 20, 26]. In these methods, the assignment of a single sample to the leaf is fuzzy or probabilistic, and gradient descent is used to optimize the tree. Most of these algorithms remain probabilistic at test time, leading to uninterpretable models as the prediction for each sample will be based on multiple leaves of the tree instead of just one.

Other advances towards the training of an oblique tree are based on constructing neural networks that reproduce decision trees [28, 48]. Yang et al. [48] use a neural network with argmax activations for the representation of classic decision tree with coordinate cuts, but they are not scalable to high-dimensional data. Lee and Jaakkola [28] use the gradient of a ReLU network with a single hidden unit at each layer and skip-connections to construct an oblique decision tree. They achieve state-of-the-art results on some datasets but have to make complete trees with a high depth, consequently having a large number of leaves.

In contrast to our method, there are other training algorithms that require the tree's structure at the beginning. Some of the works in this direction like Bennett [4] and Bertsimas and Dunn [5] use linear programming, or mixed-integer linear programming, to find an optimum global tree. Therefore, these methods are computationally expensive and not scalable. Norouzi et al. [35] derive a convex-concave upper bound on the tree's empirical loss and optimize that loss using SGD. A recent work [9] proposes tree alternating optimization, where one directly optimizes the misclassification error over separable subsets of nodes, achieving the state-of-the-art empirical performance on several datasets in a comprehensive comparison [49].

We conclude this section by connecting our splitting rule at each internal node to relevant classification algorithms [2, 23, 30, 31, 45, 51]. The two most related works are convex polytope machine (CPM) [23] and infinite support hyperplane machine (iSHM) [51], which both exploit the idea of learning a convex polytope associated decision boundary. iSHM is like a single hidden layer NN and is restricted to the binary classification task.

## 3 Convex Polytope Tree and Its Inference Algorithms

Suppose we are given the training data $(\mathbf{X}, \mathbf{Y}) = \{(\boldsymbol{x}_n, y_n)\}_{n=1}^{N}$, where pairs of $(\boldsymbol{x}_n, y_n)$ are drawn independently from an identical and unknown distribution $D$. Each $\boldsymbol{x}_n \in \mathbb{R}^d$ is a $d$-dimensional data with a corresponding label $y_n \in \mathcal{Y}$. In the classification setting, $\mathcal{Y} = \{1, \cdots, K\}$ and in regression

scenario $\mathcal{Y} = \mathbb{R}$. The aim is to learn a function $F : \mathbb{R}^d \to \mathcal{Y}$ that will perform well in predicting the label on samples from $D$. Decision tree methods construct the function $F$ by recursively partitioning the feature space to yield a number of hierarchical, disjoint regions and assign a single label (value) to each region. In what follows, we will explain how we move beyond the oblique decision trees.

## 3.1 Convex Polytope Constrained Decision Boundary

By extending the idea of disjunctive interaction (noisy-OR, probabilistic-OR) [39, 21, 50, 51] from probabilistic reasoning to the decision tree problem, we make the decision boundaries more flexible while preserving interpretability. To that end, we transform the problem of splitting at each node (right or left) to a committee of experts that make individual binary decisions ("Yes" or "No"). The committee votes "Yes" if and only if at least one expert votes "Yes," otherwise it votes "No." Thus, the final vote at each node is

$$vote = \bigvee_{i=1}^{K} vote_i,$$

where $\bigvee$ denotes the logical OR operator. We model each expert as a linear classifier who votes "Yes" with probability

$$P(vote_i = \text{"Yes"} \,|\, \{r_i, \boldsymbol{\beta}_i\}, \boldsymbol{x}) = 1 - (1 + e^{\boldsymbol{\beta}'_i \boldsymbol{x}})^{-r_i}, \tag{1}$$

where $r_i \geq 0$ and $\boldsymbol{\beta}_i \in \mathbb{R}^d$ are parameters of expert $i$. Now assuming that each expert votes independently, we can express the probability of the committee voting "Yes" as

$$P(vote = \text{"Yes"} \,|\, \{r_i, \boldsymbol{\beta}_i\}_i, \boldsymbol{x}) = 1 - \prod_{i=1}^{K}(1 - p_i) = 1 - e^{-\sum_{i=1}^{K} r_i \ln(1 + e^{\boldsymbol{\beta}'_i \boldsymbol{x}})}, \tag{2}$$

where $p_i$ is the probability of expert $i$ voting "Yes" and $K$ is the total number of experts. We can now define the split function at each node by thresholding the committee voting probability:

$$A_{left} := \{\boldsymbol{x} \,|\, \boldsymbol{x} \in \mathbb{R}^d, P(vote = \text{"Yes"} \,|\, \{r_i, \boldsymbol{\beta}_i\}_i, \boldsymbol{x}) \leq q_{thr}\} \tag{3}$$

where $A_{left}$ and $A_{right} := \mathbb{R}^d \setminus A_{left}$ are the related splits of the space.

To elaborate on the geometric shape and interpretability of the decision boundaries, consider $K = 1$ (*i.e.*, a single expert). In this scenario, the decision boundary becomes a hyperplane, which is perpendicular to $\boldsymbol{\beta}$. In fact, the probability function for each expert is based on the signed distance of $\boldsymbol{x}$ to the hyperplane perpendicular to $\boldsymbol{\beta}$. Moreover, the parameter $r$ controls how smoothly the probability transitions from 0 to 1, where a larger $r$ leads to sharper changes. This class of models with $K = 1$ and $r = 1$ are identical to oblique trees, which are interpretable. Furthermore, when $K \geq 1$, the interpretability is provided by the fact that linear classifiers and probabilistic-OR operation are both interpretable [37].

To geometrically analyze the implied decision regions, we provide the following theorem.

**Theorem 1.** *For any* $\{r_i, \boldsymbol{\beta}_i\}_{i=1}^{K}$*, such that* $r_i \in \mathbb{R}_+$ *and* $\beta_i \in \mathbb{R}^d$*, let:*

$$A_{left} = \{\boldsymbol{x} \,|\, \boldsymbol{x} \in \mathbb{R}^d, \ f(\boldsymbol{x}) \leq q_{thr}\}, \quad \text{where:} f(\boldsymbol{x}) = 1 - e^{-\sum_{i=1}^{K} r_i \ln(1 + e^{\boldsymbol{\beta}'_i \boldsymbol{x}})} \tag{4}$$

*then* $A_{left}$ *is a convex set, confined by a convex polytope.*

The proof is provided in the Appendix.

The above theorem shows that for $K \geq 1$, the decision region ($A_{left}$) is a convex set confined by a convex $K$-sided polytope. More precisely, each facet of the convex polytope is a hyperplane corresponding to an expert perpendicular to its $\boldsymbol{\beta}$. Also worth noting, an expert with a larger $r$ has more effect on the decision boundary, making sharper changes to the probability function. This can also be perceived as the value of their decision in the committee. Therefore, our method not only has a strong relationship with probabilistic-OR type models that provide interpretability for the model parameters [3] but also has decision boundaries with interpretable geometric characteristics. We propose a class of models, Convex Polytope Trees (CPT), where each tree node follows the above splitting function.

## 3.2 Gamma Process Prior

To regularize CPT and motivate simpler decision boundaries, we use a nonparamteric Bayesian shrinkage prior. Specifically, we put the gamma process prior [14, 25, 51] on the splitting function of each node in the tree. Each realization of the gamma process, consisting of countably infinite weighted atoms whose total weight is a finite gamma random variable, can be described as

$$G = \sum_{i=1}^{\infty} r_i \delta_{\boldsymbol{\beta}_i}, \text{ such that } \boldsymbol{\beta}_i \in \mathbb{R}^d, r_i \in \mathbb{R}_+ \tag{5}$$

where $\boldsymbol{\beta}_i$ represents an atom with weight $r_i$. More details about the gamma process can be found in Kingman [25] and Zhou and Carin [52]. We put the prior on the CPT by considering $\boldsymbol{\beta}_i$ and $r_i$ as the parameters of the splitting function related to Equation (2). Due to the gamma process's inherent shrinkage property, just a small finite number of experts will have non-negligible weights $r_i$ at each node. This behavior encourages the model to have simpler decision boundaries (*i.e.*, smaller number of experts or equivalently fewer polytope facets) at each node. This improves the interpretability and regularization of the model. The gamma process allows a potentially infinite number of experts at each node. For the convenience of implementation, we truncate the gamma process to a large finite number of atoms.

To further encourage simpler models at each node of the tree, we also put a shrinkage prior on $\boldsymbol{\beta}$ of each expert. In particular, we consider the prior:

$$\boldsymbol{\beta}_i \sim \prod_{j=0}^{d} \int \mathcal{N}(\beta_{ji}; 0, \sigma_{ji}^2) \text{InvGamma}(\sigma_{ji}^2; a_\beta, 1/b_{\beta i}) d\sigma_{ji}^2 \tag{6}$$

and $b_{\beta i} \sim \text{Gamma}(e_0, 1/f_0)$ which motivates sparsity due to the InvGamma distribution on the variance parameter [42, 51].

## 3.3 Training Algorithm

Finding an optimal CPT requires solving a combinatorial, non-differentiable optimization problem. This is due to the large number of possibilities that any single node can separate the data. We propose a continuous relaxation of the splitting rule at each node to alleviate this computationally challenging task. Each internal node makes probabilistic rather than deterministic decisions to send samples to its right or left branch. We set the probability of going right equal to Equation (2), or any monotonic function of it. We use this probabilistic version to train the tree in a differentiable manner. Furthermore, at the test time, we threshold the splitting functions to provide a deterministic tree. In the following subsections, we explain in detail the proposed training algorithm for the splitting parameters (Section 3.3.1) and structure of the tree (Section 3.3.2).

### 3.3.1 Learning Split Parameters

Assuming the tree structure is given, we explain how to infer the tree parameters for each task.

**Classification.** For this task, we formulate the training as an optimization problem by considering the mutual information between the two random variables $\mathbf{Y}$ (category label) and $\mathbf{L}$ (leaf id) as our objective function. This differs from the previous work on learning a classical decision tree in two main ways: 1) we develop and optimize the mutual information for a probabilistic rather than a deterministic tree, and 2) we learn the parameters of all nodes jointly rather than learning them in a greedy fashion. In terms of learning split parameters, the closest paper is Hehn et al. [20]; however, it uses an EM algorithm to learn the parameters as opposed to our novel method, where we optimize a probabilistic version of mutual information for classification. Additionally, the method in Hehn et al. [20] is only applicable to classification tasks.

We model our probabilistic tree by letting

$$\ell \sim P_{\boldsymbol{\theta}}(\ell \mid \boldsymbol{x}_n) \qquad \text{such that} \qquad \ell \in S_{leaf}, \tag{7}$$

where $S_{leaf}$ is the set of all leaf nodes and $P_{\boldsymbol{\theta}}(\ell \mid \boldsymbol{x}_n)$ is the probability of arriving at leaf $\ell$ given the sample feature $\boldsymbol{x}_n$. We assume each internal node makes decisions independent of the others and use the probabilities in Equation (2) when sending a data sample to the left or right branch. This assumption lets us derive a formula for $P_{\boldsymbol{\theta}}$ as

$$P_{\boldsymbol{\theta}}(\ell \,|\, \boldsymbol{x}_n) = \prod\nolimits_{(v, d_v)\, \in \boldsymbol{\nu}_\ell} q_{\boldsymbol{\theta}_v}(d_v \,|\, \boldsymbol{x}_n), \tag{8}$$

where

$$q_{\boldsymbol{\theta}_v}(d_v = 1 \,|\, \boldsymbol{x}_n) = 1 - q_{\boldsymbol{\theta}_v}(d_v = 0 \,|\, \boldsymbol{x}_n)$$

$$= 1 - e^{-\sum_{i=1}^{K} r_i^{(v)} \ln\left(1 + e^{\boldsymbol{\beta}_i^{(v)'} \cdot \boldsymbol{x}}\right)} \tag{9}$$

and $\boldsymbol{\nu}_\ell$ is a path from the root to leaf $\ell$ and $d_v \in \{0, 1\}$ encodes the right or left (0 or 1) direction taken at node $v$. The mutual information between $\mathbf{Y}$ and $\mathbf{L}$ can be expressed as

$$\mathcal{I}(\mathbf{Y}, \mathbf{L}) = \mathcal{H}(\mathbf{Y}) - \mathcal{H}(\mathbf{Y} \,|\, \mathbf{L}) = \mathcal{H}(\mathbf{Y}) - \sum\nolimits_{\ell \in S_{\text{leaf}}} p(\mathbf{L} = \ell)\mathcal{H}(\mathbf{Y} \,|\, \mathbf{L} = \ell), \tag{10}$$

where $\mathcal{H}(\cdot)$ indicates the entropy of a random variable, and $\mathbf{Y} \,|\, \mathbf{L}$ follows a categorical distribution. Notice that, since $\mathcal{H}(\mathbf{Y})$ does not depend on the tree parameters, to optimize mutual information, we only need to minimize $\mathcal{H}(\mathbf{Y} \,|\, \mathbf{L})$. However, the evaluation of the conditional entropy term requires knowledge of the entire data distribution, thus we can not directly optimize Equation (10).

To make the training possible, we approximate the true data distribution with the empirical one to get

$$\hat{p}(\mathbf{L} = \ell) = \sum\nolimits_{n=1}^{N} P_{\boldsymbol{\theta}}(\ell \,|\, \boldsymbol{x}_n)/N. \tag{11}$$

Denote $\mathbf{1}_{[\cdot]}$ as an indicator function. By using Bayes' rule, we derive $\hat{\boldsymbol{\pi}}^\ell = (\hat{\pi}_1^\ell, \cdots, \hat{\pi}_C^\ell)$, the estimated probability vector of the categorical distribution for $\mathbf{Y} \,|\, \mathbf{L} = \ell$, as

$$\hat{\pi}_j^\ell = \frac{\sum_n \mathbf{1}_{[y_n=j]} P_{\boldsymbol{\theta}}(\ell \,|\, \boldsymbol{x}_n)}{\sum_n P_{\boldsymbol{\theta}}(\ell \,|\, \boldsymbol{x}_n)}, \quad j = 1, \ldots, C \ . \tag{12}$$

Now by using Equation (12), we can approximate the entropy term $\mathcal{H}(\mathbf{Y} \,|\, \mathbf{L} = \ell)$ as

$$\hat{\mathcal{H}}(\mathbf{Y} \,|\, \mathbf{L} = \ell) = -\sum\nolimits_{j=1}^{C} \hat{\pi}_j^\ell \log(\hat{\pi}_j^\ell). \tag{13}$$

Therefore, we can provide an estimator for the $\mathcal{H}(\mathbf{Y} \,|\, \mathbf{L})$ as

$$\hat{\mathcal{H}}(\mathbf{Y} \,|\, \mathbf{L}) = \sum\nolimits_{\ell \in S_{\text{leaf}}} \hat{p}(\mathbf{L} = \ell)\hat{\mathcal{H}}(\mathbf{Y} \,|\, \mathbf{L} = \ell). \tag{14}$$

By minimizing $\hat{\mathcal{H}}(\mathbf{Y} \,|\, \mathbf{L})$ with respect to $\boldsymbol{\theta}$, we are in fact maximize the mutual information $\hat{\mathcal{I}}(\mathbf{Y}, \mathbf{L})$. As discussed in Section 3.2, we also regularize CPT by adding a penalty term to Equation (14). We consider the negative log probability of the gamma process prior truncated with $K$ atoms by letting

$$r_1, \ldots, r_K \overset{iid}{\sim} \text{Gamma}(\gamma_0/K, 1/c_0)$$

where $r_1, \ldots, r_K$ are the parameters of internal nodes splitting function. The penalty term for each internal node can be mathematically formulated as

$$\mathcal{L}_{reg}^{(\nu)} = \sum_{k=1}^{K} \left( -(\frac{\gamma_0}{K} - 1) \ln r_k^{(\nu)} + c_0 r_k^{(\nu)} \right) + (a_\beta + 1/2) \sum_{j=0}^{d} \sum_{k=0}^{K} [\ln(1 + \beta_{jk}^{(\nu)2}/(2b_{\beta_k}))]. \tag{15}$$

The above procedure provides a differentiable way of learning branch node parameters. At the end of the training algorithm, we also need to assign the leaf node parameters (that determine the tree's predictions at each leaf). We pass the whole training set through the tree and assign the empirical distribution of all categories at each leaf as its parameters. This way of determining the leaf parameters has been shown to achieve the highest AUC in binary-classification [15]. Algorithm 1 summarizes the training method.

**Regression.** For this task, we replace the mutual information optimization by a variance reduction criterion. To be more specific, we learn the tree parameters with

$$\underset{\boldsymbol{\theta}}{\operatorname{argmin}}\{\sum\nolimits_n P_{\boldsymbol{\theta}}(\ell \,|\, \boldsymbol{x}_n)(y_n - \hat{\mu}^\ell)^2\} \tag{16}$$

such that

---

**Algorithm 1** Stochastic gradient descent training of the tree splitting parameters for classification task.

**Input:** Data $\{(\boldsymbol{x}^{(n)}, y^{(n)})\}_{n=1:N}$, initial tree $\mathcal{T}^{(0)}$ from Topology-Leaner algorithm, maximum number of polytope sides $K$, hyper-parameters of the gamma process prior $a_0, b_0, c_0, \gamma_0$

---

    **for** number of training iterations **do**
        • Sample a batch of $m$ data samples $\{(\boldsymbol{x}_1, y_1), \ldots, (\boldsymbol{x}_m, y_m)\}$
        • Send the data samples to the current probabilistic tree to get probability of each samples being in any leaf $P_{\boldsymbol{\theta}^{(t)}}(\ell_n = \ell \,|\, \boldsymbol{x}_n)$ for all $n \in [1 : m], \ell \in S_{leaf}$ using eq. 8.
        • For each leaf, calculate the vector $\hat{\boldsymbol{\pi}}^\ell = [\hat{\pi}_1^\ell, \ldots, \hat{\pi}_C^\ell]$ which represent for the current batch what is the proportion of each data label in the leaf $\ell$ using eq. 12.
        • Estimate the entropy of the data labels $\mathbf{Y}$, conditioned on the leaf id $\mathbf{L}$ as $\hat{\mathcal{H}}(\mathbf{Y} \,|\, \mathbf{L})$ by using eq. 14.
        • Calculate the regularization term related to the gamma process prior by using eq. 15 as $\mathcal{L}_{reg} = \sum_{\nu \in S_{branch}} \mathcal{L}_{reg}^\nu$
        • Update the tree parameters by descending their stochastic gradient:

$$\nabla_{\boldsymbol{\theta}^{(t)}} \left( \hat{\mathcal{H}}(\mathbf{Y} \,|\, \mathbf{L}) + \mathcal{L}_{reg} \right)$$

    **end for**
    • Send all the data samples to the to the tree, and for each leaf set its parameter as $\hat{\boldsymbol{\pi}}^\ell$.
    • Use the threshold of 0.5 for the probabilistic decision function at each branch node to achieve a deterministic tree.

---

$$\hat{\mu}^\ell = \frac{\sum_n P_{\boldsymbol{\theta}}(\ell \,|\, \boldsymbol{x}_n) y_n}{\sum_n P_{\boldsymbol{\theta}}(\ell \,|\, \boldsymbol{x}_n)} \tag{17}$$

and the calculation of $P_{\boldsymbol{\theta}}$ is exactly the same as in the classification case. The leaf parameters are set as the mean response of the points arriving at that leaf.

Note for both scenarios, we use the threshold of 0.5 to do deterministic splits at the test time. Now that we know how to train a tree given its structure and how to use it at the test time, the next section will describe how to find an optimal topology and initial parameters.

### 3.3.2 Topology Learning

We start by assuming the tree structure to be just a root node and its two child leaf nodes. We train this tree using the algorithm explained in Section 3.3.1. After training, we split the training set to two subsets (right and left), by thresholding the assigned probability. We calculate this threshold in the classification (regression) task, which achieves maximum mutual information (variance reduction) in the deterministic tree. To be more specific, for the classification task, we set the threshold $q_{thr}$, such that it minimizes

$$\frac{n_0}{N} \cdot \mathcal{H}(\{y_{(i)}\}_{i=1}^{n_0}) + \frac{N - n_0}{N} \cdot \mathcal{H}(\{y_{(i)}\}_{i=n_0+1}^N),$$

where

$$p_{(n_0)} \le q_{thr} \le p_{(n_0+1)}.$$

The $p_{(i)}$ is the $i$'th smallest probability assigned by the root node to the data samples. We further split each child node by considering it as root and applying the above algorithm using its data samples. We stop splitting a node with very few data samples and stop growing the tree when we reach a certain predefined maximum depth.

Note, Section 3.3.1 titled as "Learning Split Parameters" (LSP) presents our algorithm for learning parameters when the structure is given. During both final refining and greedy training, LSP is used but on different structures. More specifically, at each step of greedy training, each leaf is turned into a stump (a tree with one root and two child nodes), and LSP is applied to learn only the parameters of the root node of the newly added stump, using only data points that had reached the stump's root. Then, after the greedy training, LSP is applied to the entire tree to jointly learn all the parameters, which is essentially a full tree parameter refining. We do not apply LSP to all the parameters of the tree when growing the tree during greedy training. Emphasizing that we only use LSP to train the parameter of the newly added stump, freezing the rest of the tree in a deterministic mode.

# 4 Experiments

In this section, we empirically assess the qualitative and quantitative performances of CPT on datasets from various domains. We show that CPT learns significantly smaller trees than its counterparts and makes more accurate predictions. That is partly due to the high variance of the leaf node's prediction in classical (oblique) trees, resulting from fewer data samples in each partition. However, since CPT usually has fewer leaf nodes, each partition has a significant proportion of the dataset.

Table 1: Dataset statistics

| Dataset | MNIST | SensIT | Connect4 | Letter | PDBbind | Bace | HIV |
|---|---|---|---|---|---|---|---|
| Task | Multi-class classification | | | | Regression | Binary classification | |
| Number of classes | 10 | 3 | 3 | 26 | - | 2 | 2 |
| Number of data | 70,000 | 98,528 | 67,557 | 20,000 | 11,908 | 1,513 | 41,127 |
| Feature dimension | 784 | 100 | 126 | 16 | 2,052 | 2,048 | 2,048 |

## 4.1 Synthetic Dataset

In this section we compare CPT and other tree based approaches on toy examples to better demonstrate why CPT achieves better performance. We consider a dataset of 2,000 points, as shown in Figure 1. The data samples are independent draws of the uniform distribution on a two-dimensional space $[-1, 1] \times [-1, 1]$, with the data points labeled as red if they lay between two concentric circles, and blue otherwise.

We compare CPT with LCN [28], which is a state-of-the-art oblique tree method, and CART [8], which is a canonical axis-aligned tree algorithm. Figure 1 shows the decision boundaries of all three methods. CART and LCN are trained with a maximum depth of 10, and CPT is trained with a maximum depth of 2. It is worth noting that both LCN and CART are restricted to partition the feature space into disjoint convex polytopes and assign each region to a leaf node. However, CPT does not have such a limitation, and each region can be the result of applying any set of logical operations on a set of convex polytopes. Figure 1 clearly shows that both LCN and CART need a large depth to successfully classify the data, while CPT only needs two splits. To be more specific, the AUC results for CART is $0.901$ with 9 leaves, and $0.959$ for LCN with $479$ leaves.

By contrast, CPT achieves the AUC of $0.962$ (the highest score) with just 3 leaf nodes at a maximum depth of 2. However, one limitation of our method is that it can not yet achieve competitive AUC to a multi-layer neural network.

Figure 1c shows that CPT uses a 8-sided 2D convex polytope for the first split and a 5-sided one for the second split to partition the space. This number was not fixed at training time, we only limit the maximum number of polytope sides to $K = 50$. The model owes this adaptive shrinkage to the gamma process prior, which improves the simplicity and interpretability of the model.

## 4.2 Classification and Regression

We evaluate the performance of CPT for regression, binary classification, and multi-class classification tasks. For regression and binary classification, we conduct experiments on datasets from MoleculeNet [47]. We follow the literature to construct the features [47, 27] and use the same training, validation, and testing split as Lee and Jaakkola [27]. For multi-class classification, we perform experiments on four benchmark datasets from LibSVM [10], including MNIST, Connect4, SensIT, and Letter. We employ the provided training, validation, and testing sets when available; otherwise, we create them under the criterion specified in previous works [35, 20]. The datasets statistics are summarized in Table 1.

### 4.2.1 Compared Baselines

We evaluate the performance of CPT against several state-of-the-art decision tree methods, including FTEM (Hehn et al. [20], "End-to-end learning of decision trees and forests"), TAO (Carreira-Perpinán and Tavallali [9], "Oblique decision trees trained via alternating optimization"), and LCN (Lee and Jaakkola [27], "Oblique decision trees from derivatives of ReLU networks"). We also consider several

Table 2: The performance of decision tree algorithms on regression (RMSE, lower is better) and classification (ACC or AUC, higher is better). The superscripts *, †, and ‡ is used to to show the source of the quoted results as [49], [27], and [20]

| Dataset | CART | HHCART | GUIDE | CO2 | FTEM | TAO | LCN | CPT ($K=1$) | CPT |
|---|---|---|---|---|---|---|---|---|---|
| MNIST(ACC) | 88.05±0.02* | 90.1±1.2 | 78.52±0.20* | 90±-* | 96.12±-‡ | 94.74±0.11* | 93.81±0.32 | 95.74±0.10 | **97.01±0.20** |
| SensIT (ACC) | 81.71±0.01* | - | 79.25±0.33* | 82±-* | 81.61±-‡ | 85.12±0.20* | 84.06±0.32 | 84.87±0.27 | **85.77±0.55** |
| Connect4 (ACC) | 78.29±0.21* | - | 72.01±0.36* | 78±-* | 80.51±-‡ | **81.09±0.39*** | 80.79±0.12 | 78.89±0.23 | **81.16±0.51** |
| Letter (ACC) | 86.07±0.14* | 83.1±0.3 | 82.65±0.9* | 87±-* | 86.31±0.21 | 89.15±0.88* | 89.64±0.75 | 84.13±0.32 | **90.51±0.81** |
| PDBbind (RMSE) | 1.573±0.00† | 1.530±0.00† | - | - | Not Applicable | - | 1.508±0.017† | 1.453±-0.006 | **1.413±0.004** |
| Bace (AUC) | 65.2±2.4† | 54.5±1.6† | - | - | 81.03±1.5 | 73.4±0.0† | **83.9±1.3†** | 82.06±2.4 | **84.7±1.6** |
| HIV (AUC) | 54.4±0.9† | 63.6±0.0† | - | - | 71.09±1.0 | 62.7±0.0† | **72.8±1.3†** | 71.12±2.3 | **73.1±1.4** |

Table 3: Number of leaves and (the depth of trees)

| Dataset | CART | HHCART | GUIDE | CO2 | FTEM | TAO | LCN | CPT(K=1) | CPT |
|---|---|---|---|---|---|---|---|---|---|
| MNIST | 805 (D19)* | - | **39** (D15)* | - (D14)* | 357 (D12) | 178 (**D8**)* | 65536 (D16) | 501 (D11) | 98 (**D8**) |
| SensIT | 152 (D12)* | - | **24** (D11)* | - (**D6**)* | - (D7)‡ | 69 (D7)* | 256 (D8) | 81 (D8) | 36 (**D6**) |
| Connect4 | 5744 (D33)* | - | 27 (D18)* | - (D16)* | 257 (D10) | 210 (D8)* | 65536 (D16) | 342 (D10) | **4** (**D2**) |
| Letter | 1580 (D27)* | - | 673 (D28)* | - (D12)* | 543 (D16) | 1078 (**D11**)* | 65536 (D16) | 705 (D17) | 463 (**D11**) |
| PDBbind | - | - | - | - | Not Applicable | - | 2048 (D11)† | 16 (**D4**) | 15 (**D4**) |
| Bace | - | - | - | - | 23 (D8) | - | 4096 (D12)† | **19** (D9) | 21 (**D7**) |
| HIV | - | - | - | - | 21 (D7) | - | 128 (D7)† | 25 (D7) | 24 (**D5**) |

additional baselines, including CART [8], HHCART [46], GUIDE [29], and CO2 [35]. Another related probabilistic tree work on decision trees is ANT (Tanno et al. [41], "Adaptive neural tree"), where they apply transformation at each decision tree layer to the input data, therefore they are not as interpretable as other methods. However, they achieve more than 99 percent accuracy on datasets like MNIST. We did not iclude them in the comparison table because of their non-intreprable decsion boundries. Moreover, to empirically show the importance of flexible boundaries, we also added a baseline CPT, where $K = 1$, with hyperplane cuts.

Our algorithm is implemented in PyTorch and can be trained by gradient-based methods. We use Adam [24] optimization for inferring the tree split parameters. A 10-fold cross-validation on the combined train and validation set is used to learn the hyperparameters, namely the maximum number of polytope sides, number of training epochs, learning rate, and batch-size. However, we decide the depth of the tree based on the performance of CPT on the validation set during training, which can be perceived as early stopping for trees. Following the literature, we use the Area Under the receiver operating characteristic Curve (AUC) on the test set as the evaluation metric for binary classification, accuracy (ACC) for multi-class classification, and root-mean-squared error (RMSE) for regression.

Finally, we report the average and standard error of each method's performance by repeating our experiments for 10 random seed initializations. More details about our implementation and the exact values of hyperparameters for each dataset are presented in the Appendix. Our code to reproduce the results is provided at `https://github.com/rezaarmand/Convex_Polytope_Trees`.

### 4.2.2 Experimental Results

Tables 2 and 3 present the results for a variety of decision tree based algorithms. Some results are quoted from previous works [27, 20, 49]. The depth and leaf numbers are averaged over 10 repetitions of training, and then rounded. For some large datasets, namely MNIST, SensIT, Connect4, and Letter, we fix the depth parameter as opposed to adaptively tuning it based on the validation set on each run. The reason for some missing values in Table 2 is some methods like TAO did not provide their code, so we could not provide their performance on the datasets that they had not experimented. Regarding the hyper-parameter tuning for CPT, the total number of different hyper-parameter tuning setups for each dataset was less than 25 (5*5) cases. For the baseline methods, we quote the best results tuned and reported by the authors (*e.g.*, LCN, according to their paper, does at least (3*11) hyperparameter tuning). Also, for TAO, we report the best results by the authors. For each dataset, the best result and those with no statistically significant difference (by using two sample $t$-test and $p$-value of 0.05) are highlighted.

From the results, it is evident that the added flexibility in splitting rules combined with an efficient training algorithm allows CPT to outperform the baseline algorithms. Our method achieves the state-of-the-art performance, while, notably, using significantly shallower trees. For instance, CPT obtains the best performance in Connect4 with only depth 2 and 4 leaves, while other methods need a depth of at least 8.

It also improves the regression performance on the PDBbind dataset by a large margin. Although LCN achieves competitive results in terms of accuracy on some datasets, it needs to grow the tree's size exponentially, significantly sacrificing the model interpretability. That is mainly because LCN, in contrast to our model, always learns a complete tree and generally needs to have a considerably large depth to achieve competitive results. For instance, consider its enormous size when trained on MNIST and Connect4 in Table 3.

### 4.3 Potential Negative Societal Impact

One often-neglected side effect of improving the interpretability of a machine learning model is the risk of automation bias and a false sense of trust in models. The ease of understanding the system increases the risk of misinterpretation, overtrust, and even incorrect use of the outputs. Our model's improvement of interpretability and accuracy of decision trees, which is usually employed in high stake domains, may further increase the chance of that risk.

## 5 Conclusion

We propose convex polytope trees (CPT) as a generalization to the class of oblique trees that improves their accuracy and shrinks their size, which consequently provides better predictions. CPT owes its performance to two main components: flexible decision boundaries and an efficient training algorithm. The proposed training algorithm well addresses the challenge to learn not only the parameters of the tree but also its structure. Moreover, we demonstrate the efficacy and efficiency of CPT on a variety of tasks and datasets. The empirical successes of CPT show promise for further research on other interpretable generalizations of decision boundaries. This can lead to a significant performance gain for the family of decision tree models. Another promising direction for future work is investigating the combination of CPT with various ensemble learning methods, such as boosting.

## Acknowledgements

M. Zhou acknowledges the support of NSF IIS-1812699, the APX 2019 project sponsored by the Office of the Vice President for Research at The University of Texas at Austin, and the support of a gift fund from ByteDance Inc.

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
