# Convex Polytope Trees: Appendix

## A  Proofs

In this section, we show why the splitting function at each internal node results in a convex set confined within a convex polytope. We start by proving a lemma which is needed to prove the main theorem.

**Lemma 1.** *For any $\{r_i, \boldsymbol{\beta}_i\}_{i=1}^K$, such that $r_i \in \mathbb{R}_+$ and $\beta_i \in \mathbb{R}^d$, the function:*

$$g(\boldsymbol{x}) := \sum_{i=1}^K r_i \ln\left(1 + e^{\boldsymbol{\beta}_i' \boldsymbol{x}}\right) \tag{18}$$

*is convex over its domain $\mathbb{R}^d$.*

*Proof.* Since the sum of convex functions is also convex, it suffice to show each term of $g$ is a convex function. We demonstrate this by using the following theorem: "A function is convex iff its second derivative is a positive semi-definite matrix over the domain." One can omit $r_i$'s in the following calculations because a positive scalar does not change the convexity.

The fist derivative of each term is:

$$\frac{\partial \ln\left(1 + e^{\boldsymbol{\beta}_i' \boldsymbol{x}}\right)}{\partial \boldsymbol{x}} = \frac{e^{\boldsymbol{\beta}_i' \boldsymbol{x}}}{e^{\boldsymbol{\beta}_i' \boldsymbol{x}} + 1} \boldsymbol{\beta}_i' \tag{19}$$

and by taking the derivative of the above vector, we will have:

$$\frac{\partial^2 \ln\left(1 + e^{\boldsymbol{\beta}_i' \boldsymbol{x}}\right)}{\partial \boldsymbol{x}^2} = \frac{e^{\boldsymbol{\beta}_i' \boldsymbol{x}}}{(e^{\boldsymbol{\beta}_i' \boldsymbol{x}} + 1)^2} \boldsymbol{\beta}_i \boldsymbol{\beta}_i' \tag{20}$$

where $\boldsymbol{\beta}_i \boldsymbol{\beta}_i'$ is a matrix in $\mathbb{R}^d \times \mathbb{R}^d$. Since the scalar $\frac{e^{\boldsymbol{\beta}_i' \boldsymbol{x}}}{(e^{\boldsymbol{\beta}_i' \boldsymbol{x}} + 1)^2}$ is positive for any $\boldsymbol{x}$, we just need to show the matrix $\boldsymbol{\beta}_i \boldsymbol{\beta}_i'$ is positive semi-definite. To that end, we prove for any $\boldsymbol{v} \in \mathbb{R}^d$:

$$\boldsymbol{v}'(\boldsymbol{\beta}_i \boldsymbol{\beta}_i')\boldsymbol{v} \geq 0.$$

And, that can be shown by:

$$\boldsymbol{v}'(\boldsymbol{\beta}_i \boldsymbol{\beta}_i')\boldsymbol{v} = (\boldsymbol{v}'\boldsymbol{\beta}_i) \cdot (\boldsymbol{\beta}_i'\boldsymbol{v}) = (\boldsymbol{\beta}_i'\boldsymbol{v})' \cdot (\boldsymbol{\beta}_i'\boldsymbol{v}) = \|\boldsymbol{\beta}_i'\boldsymbol{v}\|^2 \geq 0.$$

Therefore the proof of the lemma is complete.

$\square$

**Theorem 1.** *For any $\{r_i, \boldsymbol{\beta}_i\}_{i=1}^K$, such that $r_i \in \mathbb{R}_+$ and $\beta_i \in \mathbb{R}^d$, let:*

$$A_{left} = \{\boldsymbol{x} \mid \boldsymbol{x} \in \mathbb{R}^d, \ f(\boldsymbol{x}) \leq q_{thr}\}, \quad where: f(\boldsymbol{x}) = 1 - e^{-\sum_{i=1}^K r_i \ln\left(1 + e^{\boldsymbol{\beta}_i' \boldsymbol{x}}\right)} \tag{4}$$

*then $A_{left}$ is a convex set, confined by a convex polytope.*

*Proof.* We start by showing $A_{left}$ is a convex set. Note that, due to the duality

$$\boldsymbol{x} \in A_{left} \iff f(\boldsymbol{x}) \leq q_{thr} \tag{21}$$

By the definition of a convex set, we just need to prove the following:

$$\forall t \in [0,1], \forall \boldsymbol{x}_1, \boldsymbol{x}_2 \in \mathbb{R}^d \ \text{ if } \ f(\boldsymbol{x}_1), f(\boldsymbol{x}_2) \leq q_{thr} \implies f(t\boldsymbol{x}_1 + (1-t)\boldsymbol{x}_2) \leq q_{thr}. \tag{22}$$

Let $g(.)$ and $q_{thr}^*$ be:

$$g(\boldsymbol{x}) := -\ln\left(1 - f(\boldsymbol{x})\right) = \sum_{i=1}^K r_i \ln\left(1 + e^{\boldsymbol{\beta}_i' \boldsymbol{x}}\right), \quad q_{thr}^* := -\ln\left(1 - q_{thr}\right). \tag{23}$$

Since $-\ln(1-a)$ is monticaly increasing with respect to $a$, replacing $f$ by $g(.)$ and $q_{thr}$ by $q_{thr}^*$ in (22), results in a mathematically equivalent expression. Now, we can prove the new statement using Jensen's inequality. To be more specific, based on Lemma 1 ($g$ is convex) and Jensen's inequality, we have:

$$\forall t \in [0,1], \forall \boldsymbol{x}_1, \boldsymbol{x}_2 \in \mathbb{R}^d \quad g(t\boldsymbol{x}_1 + (1-t)\boldsymbol{x}_2) \leq tg(\boldsymbol{x}_1) + (1-t)g(\boldsymbol{x}_2). \quad (24)$$

So if $g(\boldsymbol{x}_1), g(\boldsymbol{x}_2) \leq q_{thr}^*$:

$$g(t\boldsymbol{x}_1 + (1-t)\boldsymbol{x}_2) \leq tg(\boldsymbol{x}_1) + (1-t)g(\boldsymbol{x}_2) \leq tq_{thr}^* + (1-t)q_{thr}^* = q_{thr}^*$$

proving $A_{left}$ is convex.

We are just remained with showing $A_{left}$ is confined within a convex polytope. This can be shown by:

$$f(\boldsymbol{x}) \leq q_{thr} \iff g(\boldsymbol{x}) \leq q_{thr}^* \implies \forall i \in [1:K], \quad r_i \ln(1 + e^{\boldsymbol{\beta}_i' \boldsymbol{x}}) \leq q_{thr}^*$$

$$\iff \forall i \in [1:K], \quad \boldsymbol{\beta}_i' \boldsymbol{x} \leq \ln(e^{\frac{q_{thr}^*}{r_i}} - 1)$$

which completes the proof. $\qquad\qquad\square$

## B  Additional details on experimental settings

As mention in the paper, we train CPT in a probabilistic manner and switch to a deterministic tree at test time. To make the transition smoother, we conduct annealing during training. To be more specific, we transform the probability function $f(\boldsymbol{x})$ at each node to $f_{\lambda_t}(\boldsymbol{x})$, where:

$$f(\boldsymbol{x}) = 1 - e^{-\sum_{i=1}^{K} r_i \ln(1 + e^{\boldsymbol{\beta}_i' \boldsymbol{x}})} \quad \text{and} \quad f_{\lambda_t}(\boldsymbol{x}) := \frac{1}{1 + (\frac{1-f(\boldsymbol{x})}{1-p_0})^{\lambda_t}} \quad (25)$$

Larger $\lambda_t$ results in a sharper change of probability from 0 to 1, and $p_0$ controls where that change happens. During training, we gradually increase $\lambda_t$ to make the gap between probabilistic and deterministic tree progressively smaller. We also learn $p_0$ like other parameters of the model using SGD. Notice, the change of $f$ to $f_{\lambda_t}$ keeps the mathematical and geometrical interpretation of CPT intact. That is because any thresholding of $f_{\lambda_t}$ has an equivalent counterpart for $f$ since $f$ and $f_{\lambda_t}$ are strictly monotonic function of each other.

Note, the main goal of our comparison in Table 2 is reporting the highest possible value a method can achieve regardless of the size. This has significant importance in the decision tree literature, as they are often undermined since their accuracy is lower than the NN counterparts, and it is important to push the accuracy limit of these classes of methods because they provide interpretability which is missing with NNs.

For the comparison, we tried our best to state the best performance reported for each method and if there were no results on a specific dataset and code was available, we trained their model up to the depth of 16. For the FTEM method, the reported numbers are based on the author's best hyper-parameter tuning. They varied the maximum depth from 2-18 with a step size of 2. These numbers were only available for MNIST, Connect4, and SensIT. For other classification results, we trained their model and reported the results. However, their method was not applicable to regression, so we did not report the results.

For TAO, the code is not available. However, the authors of TAO have written a follow-up paper [50] comparing old and new decision trees. We use the results in this paper to report the performance of TAO on MNIST, Connect4, SensIT, and Letter. They did not provide all the details of how they varied the maximum length, but for some methods, they varied the depth up to 30. And for the results on HIV and Bace, we relied on the reported numbers in LCN (Lee and Jaakola, 2020), where they vary the depth of the tree in the interval [2, 12].

For the LCN method on HIV, Bace, and PDBbind, we used their reported results in which they varied the depth from 2, 3, .., 12. It is worth noting that LCN does not learn the tree structure, and it always learns a full binary tree.

We did not compare our method with soft trees (soft at test time) that are often not considered interpretable. For this reason, they have not been previously compared to the deterministic trees in

the literature. However, it is worth mentioning that CPT results for the soft version considerably improve if we use it as a soft tree at the test time. For instance on the Bace and HIV datasets, we can achieve 1-2 percentage increase in AUC if we use the soft version. The only closely related method is FTEM. During the training period, they are also a probabilistic tree and use the thresholded soft tree at the test time, which we do compare against in our experiments.

We performed the classification and regression experiments on a laptop with a 2.5 GHz 6-Core Intel Core i7 CPU and 16 GB of RAM.