# OpenReview forum: "Convex Polytope Trees"
_NeurIPS.cc/2021/Conference — NeurIPS 2021 Poster_

### Official Review · Reviewer_HSsp · 2021-07-13

**Rating:** 3
**Confidence:** 4

**Summary:**

The paper proposes a  new convex polytope trees to build the family of decision trees.

**Limitations And Societal Impact:**

1, The paper claims that existing tree methods produce large size tree and the  computational complexity is too high. But what is the resulting tree size and time cost of the proposed tree method? what is the comparisons with existing tree methods in terms of tree size and  training and testing time cost? There are no theory analysis and no experiment report.
2, what is vote_i?  why the probability of a linear classifier who votes "YES" can be shown as in Eq.(1)? I do not see any reasons.
how do define expert i? what is q_thr?
3, Theorem 1 shows that A_left is a convex set, confined by a convex polytope. what is the role of Theorem 1? why convex set is good? I do not see any benefit. Why use gamma process  prior?
4, In MNIST data set, in table 1, the proposed method achieves 97.01, the performance is significantly below SODA method.

**Main Review:**

The paper proposes a  new convex polytope trees to build the family of decision trees, and combine VAEs for  generative modeling issues. The idea seems fine, but there are some major concerns.
1, The paper claims that existing tree methods produce large size tree and the  computational complexity is too high. But what is the resulting tree size and time cost of the proposed tree method? what is the comparisons with existing tree methods in terms of tree size and  training and testing time cost? There are no theory analysis and no experiment report.
2, what is vote_i?  why the probability of a linear classifier who votes "YES" can be shown as in Eq.(1)? I do not see any reasons.
how do define expert i? what is q_thr?
3, Theorem 1 shows that A_left is a convex set, confined by a convex polytope. what is the role of Theorem 1? why convex set is good? I do not see any benefit. Why use gamma process  prior?
4, In MNIST data set, in table 1, the proposed method achieves 97.01, the performance is significantly below SODA method.

**Time Spent Reviewing:**

0.5

---

> ### Author Response · Authors · 2021-08-10
> **Our Thanks and Reply to Reviewer HSsp**
>
> We appreciate and thank your thoughtful and detailed comments, we believe addressing your points will significantly improve our work. Please find in the following our responses.
>
> 1- We will further clarify in the paper that we *only* claim lower computational complexity of CPT over existing methods for the generative modeling task (when combined with VAEs) and not necessarily when used for canonical tasks. This is because the main computational bottleneck in the generative task are the VAEs, and the number of VAEs equals the number of leaves. Thus, the fewer number of leaves in CPT provides a significant computational advantage over other methods.
>
>
> 2- We model the decision of sending a node to the right or left as the aggregate opinion of a committee of experts who vote yes or no. This aggregate (final decision) is yes if at least one of the experts votes yes, similar to a logical OR operation. We will modify lines 119-121 to better define vote_i.
> Equation (1) is a choice (specific to our method) rather than a derived equation. This equation enables us to achieve two things: 1) {beta’_i x} measures the distance from x to the hyper-plane related to beta_i. Without r_i, equation 1 is the canonical linear classifier softmax. 2) Adding r_i allows the model to have the capacity to prioritize some hyperplanes over others.
> q_th is an arbitrary number used as a threshold at each node to convert the soft tree to a deterministic tree.
>
>
>
> 3- It is arguably important to study and understand the geometry of predictive models' decision boundaries. Theorem 1 is solely to provide an understanding of the geometry of the decision boundaries in CPT.
>
>
> We will clarify in the paper that we do not have a preference over convex or non-convex, and the purpose of the theorem is to provide a better understanding of the geometry of the decision boundaries for future readers/researchers.
>
>
> We use gamma process prior because they naturally fit within our problem formulation, and they are a well-accepted prior in non-parametric Bayes literature when we do not want to limit the number of polytope facets, but we want to motivate the model to have a small number of polytope facets. We provide intuition in the Appendix, and we will expand the current explanations.
>
>
>
> 4- Classification accuracy on MNIST is significantly below SOTA methods:
> To the best of our knowledge, 97.01 is SOTA on MNIST among decision trees literature. However, it is definitely far from SOTA when NNs are used, but they are not interpretable.

---

> ### Author Response · Authors · 2021-08-30
> **Reply to Reviewer HSsp**
>
> Dear reviewer,
>
> We are grateful for your detailed comments. We would be thankful if you could please let us know whether our response has addressed all of your concerns? If not, please point us to the questions we did not address adequately, and we would be happy to provide additional explanations or experiments.

---

### Official Review · Reviewer_5qk9 · 2021-07-16

**Rating:** 6
**Confidence:** 5

**Summary:**

The paper proposes a method for learning decision trees with convex splits. It does so by replacing the common linear splits by a noisy-OR operation over a learned set of linear splits with a shrinkage prior to limit the size of this set.
It further shows how to apply the tree learning procedure to tackle three downstream tasks (classification, regression and density estimation with VAEs).
An empirically evaluation is also provided comparing the obtained method with several baselines, showing that it generally allows to obtain good (or better) performance with a smaller amount of nodes.

**Limitations And Societal Impact:**

Yes.

**Main Review:**

The proposed CPT could be viewed as learning trees with oblique splits but where some branches are pruned. Indeed, a pruned tree can be expressed in a more compact form by replacing a particular branch by a single node with a convex split defined by the hyper-planes of the nodes of that branch.
This is because in trees with oblique splits a branch can be interpreted as a sequence of AND decisions, and some of the sequences define a convex set.
The advantage of the proposed approach is in automatically choosing a single branch among the multitude of combinations of splits, which makes the tree more interpretable without having to prune and rearrange it afterwards.
**Directly learning convex splits is an interesting idea and novel in the context of decision trees**, to the best of my knowledge.

The paper is generally well written, with assumptions and method derivations generally well reported. However, **two key points need clarification**:

1. It is not clear whether while the tree is grown greedily, the parameters of all the nodes are updated jointly every time the tree is expanded. This confusion stems from an apparent contradiction between paragraph 231-234, stating that no parameter refinement is performed, and what previously stated in line 166-167, that the parameters of all nodes are learned jointly.
This is a key point to clarify, as it would be a limitation of the work. If the tree is grown greedily, it is not optimal, as opposed to other soft tree learning techniques or MIP-based methods.

2. The specific problem for learning the set of hyper-planes at each node is not sufficiently described. I understand that the Gamma process is a common shrinkage prior in the Bayesian community, but it is new in the tree learning context. It would be valuable to report more information (even in the appendix) on the convex split optimization, such as how the prior set of hyper-planes is chosen and how the Gamma process works.
Moreover, the reasons for choosing the specific form for the expert probabilities (Equation 1) is never discussed.

**The empirical evaluation is the weakest point** of the work. The application of the method to three different tasks (classification, regression and density estimation) is valuable, however the comparison with state-of-the-art methods should be strengthened and some important details of the setup are missing:

3. In order for the comparison to be fair the total number of parameters of the trees should be reported, and not only the number of nodes. The number of nodes is indeed a good proxy measure for the interpretability of the model, but it does not account for the total number of hyper-planes needed for good accuracy.

4. In this sense, the considered methods should be compared in terms of accuracy (or the other prediction scores) for same (or similar) number of hyper-planes. However this does not seem the case in the current evaluation.

5. The values or ranges of the hyper-parameters of the baselines are not reported, in particular the maximal tree depth.

6. No comparison is provided with soft tree baselines, which are the methods most closely related to this work.

Finally, it is claimed that an advantage of the method w.r.t. existing techniques is in not having soft decisions, so points traverse the tree deterministically. However this is true only at test/inference time, by thresholding the splits, hence the same reasoning can be applied to other soft tree techniques.

### UPDATE
The authors addressed well my concerns. In particular, they provided tree sizes in number of hyper-planes, and even though they do not compare models' accuracy for a same tree size, I am convinced by their argument that it is valuable to compare models in terms of best achievable performance. They also clarified the several points raised in the different reviews.

Overall, I think that the contributions are new in the tree learning literature and would be interesting to this community, even though they principally leverage existing tools in Bayesian learning.
I hence increased my rating to 6, with the hope that the authors will revise the paper according to all our suggestions.

**Time Spent Reviewing:**

6

---

> ### Author Response · Authors · 2021-08-10
> **Our Thanks and Reply to Reviewer 5qk9**
>
> We sincerely appreciate the thoroughness and detail that went into your review. Thank you for all of the constructive feedback! We believe that your feedback enabled us to significantly clarify/improve the paper. Please find our response below:
>
> 1- Thank you for bringing this to our attention. A better explanation is needed regarding how the greedy training is done. We will edit the paper to clarify them.
> Section 3.2.1 "Learning Split Parameters" (LSP)  presents our algorithm for learning parameters *when* the structure is given. During both final refining and greedy training, LSP is used but on different structures. More specifically, at each step of greedy training, each leaf is turned into a stump (a tree with one root and two child nodes), and LSP is applied to learn only the parameters of the root node of the newly added stump, using only data points that had reached the stump's root. Then, after the greedy training, LSP is applied to the entire tree to jointly learn all the parameters, which is essentially a full tree parameter refining.
>
> *Explanation regarding paragraph 231-234 and lines 166-167:
>
> In 231-234, we wanted to say that we do not apply LSP to all the parameters of the tree when growing the tree during greedy training. Emphasizing that we only use LSP to train the parameter of the newly added stump, freezing the rest of the tree in determinist mode. However, in lines 166-167, we wanted to refer to the full refining of the tree when all the parameters are trained.
>
>
> 2.1- Gamma process details:
>
> We agree that a more comprehensive explanation about the Gamma process prior needs to be added to the paper. We will improve and add more details to section A of the appendix.
> We set $\gamma_0 = c_0 =1$ and a_{\beta} = b_{\beta k} = 1e-6 for all the experiments and use it to shrink the number of convex polytope facets and will explain in detail how it works in the paper.
>
> 2.2- Reasons for choosing equation 1:
>
> Thank you again for bringing up another very helpful point that will benefit from more explanations. Equation 1 enables us to achieve two things: 1) since beta_i’ x measures the distance from x to the hyper-plane related to beta_i, we set P(yes) accordingly. In other words, Without r_i, equation 1, is the canonical linear classifier softmax. 2) Adding r_i allows the model to have the capacity to prioritize some hyperlanes over others.
>
>
> 3- This is a valid remark, we will add the numbers to the paper.
>
> However, we need to discuss the following point: Many of the current tree methods do not see performance gains even when very large depths (e.g., 100) are allowed. So we sought to compare each model's *best* attainable results by doing an exhaustive search (or use results from people who did search a wide range) of parameters and depths.
>
> CPT has 733 and 75 hyperplanes for MNIST and Connect4 datasets. The second best model, in terms of accuracy, on MNIST is FTEM with 619 hyperplanes. On connect4, TAO has the second best performance and has more than 210 (since it has at least 210 leaves, we do not have the code to report the exact number of hyperplanes for TAO).
>
>
> 4 and 5) We will clarify in the paper that the main goal of our comparison table [1] was reporting the highest possible value a method can achieve regardless of the size. This has significant importance in the decision tree literature, as they are often undermined since their accuracy is lower than the NN counterparts, and it is important to push the accuracy limit of these classes of methods because they provide interpretability which is missing with NNs.
>
> With that being said, we tried our best to state the best performance reported for each method and if there were no results on a specific dataset and code was available, we trained their model up to the depth 16. We will also add the following specific details to the Appendix:
>
> For the FTEM method, the reported numbers are based on the author’s best hyper-parameter tuning. They varied the maximum depth from 2-18 with a step size of 2. These numbers were only available for MNIST, Connect4, and SensIT. For other classification results, we trained their model and reported the results. However, their method was not applicable to regression, so we did not report the results.
>
> For TAO, the code is not available. However, the authors of TAO have written a follow-up paper (Zharmagambetov et. al, 2019) comparing old and new decision trees. We use the results in this paper to report the performance of TAO on MNIST, Connect4, SensIT, and Letter. They did not provide all the details of how they varied the maximum length, but for some methods, they varied the depth up to 30. And for the results on HIV and Bace, we relied on the reported numbers in LCN (Lee and Jaakola, 2020), where they vary the depth of the tree in the interval [2, 12].
>
> For the LCN method on HIV, Bace, and PDBbind, we used their reported results in which they varied the depth from {2, 3, .., 12}.  It is worth noting that LCN does not learn the tree structure, and it always learns a full binary tree.
>
>
> *Comparison is provided with soft tree baselines:
>
> We will clarify in the paper that soft trees (soft at test time) are not considered interpretable. For this reason, they have not been previously compared to the deterministic trees in the literature. However, it is worth mentioning that CPT results for the soft version considerably improve if we use it as a soft tree at the test time. The only closely related method is FTEM (Hehn et. al, 2019). During the training period, they are also a probabilistic tree and use the thresholded soft tree at the test time, which we do compare against in our experiments.

---

> ### Author Response · Authors · 2021-08-30
> **Reply to Reviewer 5qk9**
>
> Dear reviewer,
>
> We are thankful for your detailed and thoughtful comments. We would be grateful if you could please let us know whether our response has addressed all of your concerns? If not, please point us to the questions we did not address adequately, and we would be happy to provide additional explanations or experiments.

---

> > ### Comment · Reviewer_5qk9 · 2021-08-31
> > **updated review**
> >
> > Dear authors,
> > thank you for your patience and please find an update in my review with the reasons that brought me to increase my rating to 6.

---

> > > ### Author Response · Authors · 2021-08-31
> > > **Re: updated review**
> > >
> > > Dear Reviewer,
> > >
> > > We sincerely appreciate your recognizing our contributions and giving us additional feedback. There are many insightful comments and suggestions, which we will take into careful consideration when revising our paper.

---

### Official Review · Reviewer_54hJ · 2021-07-16

**Rating:** 6
**Confidence:** 3

**Summary:**

Authors develop a generalization of oblique trees, convex polytope trees. They are able to implement them in a differentiable, end-to-end, scalable way. In their formulation. Authors compare against competitive baselines and obtain state of the art results

**Limitations And Societal Impact:**

Regarding societal impacts and limitations" I believe the commentary on interpretability should be improved. Currently, the loss of interpretability due to a more complex region are not being put into the interpretability discussion (which only focuses on the  regularization aspects, which are nonetheless appreciated)

**Main Review:**

I think it is a strong paper since
a)It is well written
b) it addresses a relevant topic: trees are important objects and there are still a growing literature in how to learn trees using more modern paradigms (e.g. differentiable programing).
c)The methodology is original and sensible.
d)The experimental validation is quite strong; beyond standard tree formulations, authors also compare against the recent LCN [1], producing better results. Then, this paper can be understood as improving upon [1]. Although the improvement may be a bit marginal (it goes on the same direction, but extending the class of sets used at each split and introducing regularization), it is still significant and praise deserving.
e)Figure 2 is really nice and striking. I would even recommend to move it upwards so the point is made earlier in the paper.


I have some concerns, that I would like to be addressed
1)Interpretability: One of the main reasons for why trees are popular is because the original regions X>=a were easy to interpret. I am not surprised that the increased expressive power of oblique or polytope leafs would lead to improves in accuracy and shortening the depth of the tree. However, I believe this methodology has the drawback of being inherently less interpretable. For example, for a practitioner trying to use trees for a diagnosis. Having polytope regions would appear much more black-box-ish than the original trees. I am surprised tthat the authors promote the idea that their method is more interpretable than alternatives. While some of the aspects are more interpretable, I think having a polytope in the split hurts interpretability and this is not being commented. Authors should comment more about interpretability and the possible tradeoff/relation between complexity of each split, depth of the resulting tree, and capabilities to interpret. Perhaps trying to incorporate elements from [2] into the discussion would help


2)Is the method for learning split parameters new? how does it compare to [1] (and all previous work)? If  it is new, authors should highlight this novelty. If not, authors should cite the original reference.
3)Authors talk about "Robustness" in predictions. But I don't see any formal treatment of robustness in the paper. Therefore, the claim appears unsubstantiated. I would appreciate that the authors make more explicit what they mean by robustness here or otherwise change the word.


[1]https://arxiv.org/abs/1909.13488
[2]https://www.nature.com/articles/s42256-019-0048-x

**Time Spent Reviewing:**

5

---

> ### Author Response · Authors · 2021-08-10
> **Our Thanks and Reply to Reviewer 54hJ**
>
> Thank you for all of the constructive comments. We sincerely appreciate the thoroughness and detail that went into your review.
>
>
> e) Thank you for letting us know that this figure is helpful for the readers, we will move the figure earlier.
>
>
> 1- Thanks for raising this important concern about interpretability and providing the very interesting related paper [2]!
> We agree that our explanations can imply that we are looking to improve interpretability, and we need to discuss the trade-offs. However, our main goal/claim was not to improve interpretability. We were mainly concerned about pushing accuracy limits while staying in the class of interpretable models. In doing that we offer a trade-off: “shallow trees with noisy-OR decision boundaries and higher accuracy” as an alternative to “deep trees with X>=a decision boundaries”. As you mentioned, choosing which model to use depends on the practitioner and the application at hand. We will add the following to the paper, explaining the above in more details:
> “””
> Neural networks, with more than two layers, are considered black-box models (not in the class of interpretable models). However, even with larger depth, lots of leaves, or complex (but interpretable) decision boundaries, decision trees would still remain in the class of interpretable models. Our proposed convex-polytope tree offers higher accuracy than previously studied tree models and remains interpretable (Noisy-OR decision boundaries are widely considered interpretable).
> While it is hard to objectively compare the interpretability \footnote{“Interpretability is a domain-specific notion . . . so there cannot be an all-purpose definition”~\cite{rudin2019stop}.} of convex polytope trees vs. oblique trees, they offer an alternative where \emph{ease of interpretability of the decision boundaries at each node} can be traded for \emph{shallower trees with significantly fewer leaves}. Choosing one over the other is often application dependent, with the added advantage that convex polytope trees often offer higher accuracy.
> “””
>
> 2- We agree that we should have more clearly explained how our parameter learning compares to others. We will expand lines 164-167 to outline the novelty and differences to other works more clearly. As you mentioned, in terms of learning split parameters, the closest paper is [1]; however, [1] uses an EM algorithm to learn the parameters as opposed to our novel method, where we optimize a probabilistic version of mutual information for classification. Additionally, the method in [1] is only applicable to classification tasks.
>
>
> 3- We agree that our use of “Robustness” may be confusing; we will remove it from the paper. We intended to convey that: Most decision tree algorithms make predictions on each region only based on the training samples within that region. This can lead to higher uncertainty if the region/leaf has a small number of training points.
> For instance, in the classification task, most methods calculate P(y|X_test_i) based on the proportion of training samples with class y within only the region/leaf of X_test_i. Since other decision tree methods usually have more leaves/regions, inference for each data point would be based on fewer data points within that leaf. This can lead to higher uncertainty. We will make this clarification in the paper too.
>
> *Thank you for bringing up the societal impacts and limitations concern. We agree that this is a critical point to consider for anyone using our method in practice. We are going to elaborate on the trade-offs between accuracy, interpretability, and depths of tree models (similar to our response to your comment 1)

---

### Official Review · Reviewer_pRjS · 2021-07-19

**Rating:** 8
**Confidence:** 4

**Summary:**

The authors propose an alternative decision tree model, where the splitting criteria/decision boundary are convex polytopes instead of axis aligned halfplanes or even more general oblique hyperplanes.

One advantage of this is that it gives us another mechanism to add model capacity (for traditional decision trees, model capacity is typically controlled by adding trees or increasing depth). The proposed approach adds capacity by making the splitting criterion/decision boundary more flexible.

As they show via experiments, this leads to requiring fewer trees/leaf nodes, which could potentially lead to more interpretable models.

In addition, their technique can use SGD based techniques allowing it to be used in conjunction with other (differentiable) downstream techniques, not to mention potentially being used for larger datasets.


**Limitations And Societal Impact:**

This seems to be a useful technique both for improving accuracy (and potentially) for improving interpretability.

**Main Review:**

The paper is well written and clear. The technique presented seems to be novel, and applicable to a variety of use cases that decision trees have not been traditionally used for. And the results presented show that the technique is effective.

Overall, this seems like a solid paper. One downside though is that the need for fewer leaves is motivated by the need for interpretability. However, the experiments are primarily about accuracy (the assumption seems to be that smaller number of leaves will automatically lead to better interpretability, but that is not shown experimentally). Figure 1 seems to indicate that the VAE use case will result in splitting up the density region into interesting/interpretable regions, but it is not clear if that is borne out by the experiments.


**Time Spent Reviewing:**

4

---

> ### Author Response · Authors · 2021-08-10
> **Our Thanks and Reply to Reviewer pRjS**
>
> We sincerely appreciate your time, feedback, and recognition of the significance of our work!
>
> *Number of leaves will automatically lead to better interpretability:
> 	We agree that our explanations can imply that we are looking to improve interpretability. However, our main goal was to push accuracy limits while staying in the class of interpretable models. We make sure to fully clarify this point in the paper.
>
> *Experiment needed regarding CPT-VAE splitting up the density region into interesting/interpretable regions:
> 	We agree, and thanks for bringing this up. We do have some experiments (Figure 3) in the appendix, but we failed to clearly highlight them in the main paper. For future readers, we will make a note in section 4.3. In Figure 3, we present the partitions’ contingency matrix. The figure shows that there is mostly a single dominant value per column and most leaves are responsible for generating a significant proportion of at least one label. This indicates that leaf ID, to some extent, captures semantic information.

---

### Decision · Program_Chairs · 2021-09-27

**Decision:**

Accept (Poster)

**Comment:**

Reviewers were very split about this paper, so much so that they did not come to a consensus. After looking through the paper and discussion I have decided to vote to accept for the following reasons: (i) usefulness: trees are an important class of ML models due to their speed, interpretability, accuracy, and ability to handle data with different scales. This work is able to maintain these properties while improving accuracy over recent work; (ii) novelty: while convex polytopes have been used as decision boundaries before it is non-trivial to integrate them into a learning procedure for trees: the proposed approach frames it in a principled way as a maximization of mutual information; (iii) clarity: the paper clearly presents the ideas it introduces.
Alongside the recommendations made by reviewers (which the authors should carefully take into account when preparing the camera-ready version) I would urge the authors to consider the following things: (a) remove generative modeling: I don't think the generative modelling part of the paper adds much to the paper and I would consider removing this part and retitling the paper "Convex Polytope Trees". I think most readers interested in decision trees will not be interested in this aspect, and it detracts from the most important contributions (the new splitting function, the fully differentiable training). Instead I would emphasize (possibly with an instructive diagram) that your approach can be used as a drop-in NN layer (because it is fully differentiable). This will get the point across without distracting reviewers with additional experiments; (b) details on the gamma process prior: it would be nice to see an experiment of the effect of the gamma process prior. At the moment this is mentioned very briefly in the paper, but it would be very instructive to see the role it plays in the trees that are built (e.g., how do the trees change without this prior?); (c) algorithm: include a more succinct version of algorithm 1 in the main body of the paper, this will make the training of the method much clearer to the reader; (d) instructive figures: it would be nice to have another instructive figure explaining how the splits in this work differ from classical and modern tree methods; (e) additional baseline: it would be great for the authors to compare against Tanno, R., Arulkumaran, K., Alexander, D., Criminisi, A., and Nori, A. Adaptive neural trees. In Proc. of ICML, 2019 as it performs very competitively in prior work (I am not an author of this work). If the authors are able to make these changes and the ones mentioned by the reviewers it will make a good paper even better, and a nice contribution to the conference.